# Characterization of the Technofunctional Properties and Three-Dimensional Structure Prediction of 11S Globulins from Amaranth (*Amaranthus hypochondriacus* L.) Seeds

**DOI:** 10.3390/foods12030461

**Published:** 2023-01-19

**Authors:** Jorge Aguilar-Padilla, Sara Centeno-Leija, Esaú Bojórquez-Velázquez, José M. Elizalde-Contreras, Eliel Ruiz-May, Hugo Serrano-Posada, Juan Alberto Osuna-Castro

**Affiliations:** 1Facultad de Ciencias Químicas, Universidad de Colima, Carr. Colima-Coquimatlán km. 9, Coquimatlán 28400, Colima, Mexico; 2Facultad de Ciencias Biológicas y Agropecuarias, Universidad de Colima, Carr. Colima-Manzanillo km. 40, Tecomàn 28100, Colima, Mexico; 3Consejo Nacional de Ciencia y Tecnología, Laboratorio de Biología Sintética, Estructural y Molecular, Laboratorio de Agrobiotecnología, Tecnoparque CLQ, Universidad de Colima, Carretera Los Limones-Loma de Juárez, Colima 28629, Colima, Mexico; 4Red de Estudios Moleculares Avanzados, Instituto de Ecología A.C., Cluster BioMimic®, Carretera Antigua a Coatepec 351, El Haya, Xalapa 91073, Veracruz, Mexico

**Keywords:** mass spectrometry analysis, assembly capacity, physicochemical and functional properties, bioinformatic analysis

## Abstract

Amaranth 11S globulins (Ah11Sn) are an excellent source of essential amino acids; however, there have been no investigations on the characterization of their techno-functional properties at different pH conditions and NaCl concentrations, which are necessary for food formulations. In this work, we report a new two-step purification method for native Ah11Sn with purity levels of ~95%. LC–MS/MS analysis revealed the presence of three different Ah11Sn paralogs named Ah11SB, A11SC, and Ah11SHMW, and their structures were predicted with Alphafold2. We carried out an experimental evaluation of Ah11Sn surface hydrophobicity, solubility, emulsifying properties, and assembly capacity to provide an alternative application of these proteins in food formulations. Ah11Sn showed good surface hydrophobicity, solubility, and emulsifying properties at pH values of 2 and 3. However, the emulsions became unstable at 60 min. The assembly capacity of Ah11Sn evaluated by DLS analysis showed mainly the trimeric assembly (~150–170 kDa). This information is beneficial to exploit and utilize Ah11Sn rationally in food systems.

## 1. Introduction

Amaranth (*Amaranthus hypochondriacus* L.) is a pseudocereal considered a promising alternative crop due to its nutritional and agronomical characteristics. Amaranth seeds have been cultivated since pre-Columbian times in Mesoamerica and are nutritionally superior in protein content (~18%) to widely consumed conventional cereals (7–10%). Amaranth seeds have a high content of seed storage proteins (SSPs), with a well-balanced amino acid composition close to the optimum required in the human diet established by the FAO; additionally, SSPs are rich in active peptides with diverse functions [1,2,3,4], which make amaranth seeds a desirable source of nutritious protein for human consumption. Additionally, amaranth plants are tolerant to several abiotic stresses, such as high temperatures, salinity, and water deficit [5,6,7].

Saline-soluble globulins are one of the significant SSP fractions in amaranth seeds [4]. Based on their sedimentation coefficient, globulins are classified as 7S globulins or vicilins and 11S globulins or legumins. In general, 11S globulins have hexameric three-dimensional (3D) structures of ~275–450 kDa composed of ~50–60 kDa monomers consisting of α (~30–40 kDa) and β (~20–25 kDa) subunits, linked by a highly conserved disulfide bond [8,9].

The application of 11S globulins in the food industry is determined by their physicochemical and functional properties, from which surface hydrophobicity, solubility, emulsifying properties, foam, and gel-forming capacity are among the most appreciated [10]. Nevertheless, the physicochemical and functional properties of purified native amaranth 11S globulins (Ah11Sn) are not well known. This information is of utmost importance to support the use of individual amaranth proteins in commercial food products.

The purification and characterization of the solubility of Ah11Sn under different pH conditions, heat-induced gelation, and their *in silico* surface hydrophobicity were previously reported [11]. However, their emulsifying properties, as well as their surface hydrophobicity and assembly capacity, have not been experimentally characterized. In this work, we report a new two-step purification method that yields Ah11Sn at up to 95% purity. The surface hydrophobicity, solubility, emulsifying properties, and assembly capacity were experimentally characterized at different pH values and NaCl concentrations, conditions of interest for food processing. Additionally, the presence of three different monomers of Ah11Sn was identified by LC–MS/MS, and their three-dimensional (3D) structures were modeled using the new-generation protein structure predictor software AlphaFold2 [12]. The experimental characterization of the techno-functional properties of proteins supported by 3D structure modeling using bioinformatics software sheds light on the knowledge of the Ah11sn structure–function relationship and their behavior in different conditions of interest for the food industry.

## 2. Materials and methods

### 2.1. Materials and Chemical Reactants

The mature seeds of *A. hypochondriacus* used in this work were donated by Ana Paulina Barba de la Rosa, Ph.D., from IPICyT (Instituto Potosino de Investigación Científica y Tecnológica, San Luis Potosí, México). NaCl, K_2_HPO_4_, EDTA, glycine, citric acid, Na_2_HPO_4_, Trizma base, 8-anilino-1-naphthalene sulfonate (ANS), and HCl were purchased from Sigma–Aldrich (St. Louis, MO, USA).

### 2.2. Purification of 11S Globulins from Amaranth Seeds

Amaranth 11S globulins (Ah11Sn) were extracted from mature amaranth seeds following the method described previously [11] with some modifications. The crude globulin extract was precipitated at 60% ammonium sulfate saturation, dialyzed (16 h, 4 °C) against distilled water, and lyophilized. Ah11Sn was purified by a first chromatographic step using a 2 cm × 13 cm DEAE-Sepharose column (GE Healthcare, Piscataway, NJ, USA) equilibrated with buffer A (50 mM NaCl, 10 mM K_2_HPO_4_, 1 mM EDTA, pH 7.6) and eluted with buffer B (buffer A + 0.4 M NaCl). Fractions of 10 mL were collected and analyzed by absorbance at 280 nm (A_280_) using a Plus spectrophotometer (Eppendorf, Germany), dialyzed against distilled water, and lyophilized. A second chromatographic step was carried out using a molecular size exclusion (SEC) 2.5 cm × 68 cm Sephacryl S-300 column (GE Healthcare, Piscataway, NJ, USA) using buffer B as a mobile phase and previously calibrated with a gel filtration standard kit (Bio-Rad, Laboratories, Hercules, CA, USA). Fractions of 2.5 mL were collected and analyzed by A_280_, dialyzed against distilled water, and lyophilized. The purity of the samples was analyzed by 12% SDS–PAGE [13] under reducing conditions with 2-mercaptoethanol; proteins were stained with 0.05% Coomassie Brilliant Blue R-250 in methanol:acetic acid:water (40:10:50 v%). Gels were analyzed by densitometry with ImageJ software [14].

### 2.3. Protein Quantification

Protein quantification was measured by the Bradford method [15] with a protein assay kit (Bio-Rad, Laboratories, Hercules, CA, USA) using BSA as a standard.

### 2.4. Protein in-Gel Digestion, Mass Spectrometry Analysis, and Protein Identification

The gel bands were dissected manually and destained with 2.5 mM NH_4_HCO_3_ (AB) in 50% acetonitrile (ACN) and then dehydrated with 100 μL of 100% ACN. The proteins were reduced with 20 μL of 10 mM DTT in 50 mM AB and incubated for 45 min at 56 °C. After that, the samples were cooled at 25 °C, alkylated with 100 mM iodoacetamide in 50 mM AB, and incubated in the dark for 30 min. The gel cubes were washed with 100 μL of 100% ACN for 5 min and rinsed with 100 μL of 5 mM AB for 5 min. Then, the samples were rewashed with 100 μL of 100% ACN for 5 min and dried with a CentriVap (Labconco, Kansas, MO, USA) for 5 min. Then, the samples were rehydrated with 10 μL of digestion solution containing 12.5 ng/μL mass spectrometry grade Trypsin Gold (Promega, Madison, WI, USA) in 5 mM AB. The reaction was carried out in a water bath at 37 °C overnight and stopped at −80 °C. The resulting peptides were extracted three times with 30 μL of 50% ACN with 5% formic acid by centrifugation at 1000× *g* for 30 s. Finally, samples were desalted with ZipTip-μC18 tips (Merck Millipore, Darmstadt, Germany) and dried using a CentriVap (Labconco).

The LC–MS/MS analysis was carried out as previously reported [16]. Raw data were analyzed with Proteome Discoverer 2.4 (Thermo Fisher Scientific Inc., Waltham, MA, USA) and subsequent searches using the Mascot, SQUEST HT, and MS Amanda engine algorithms against *A. hypochondriacus* v1.0 and v2.1, previously reported [17,18] and obtained from Phytozome 12 (https://data.jgi.doe.gov/refine-download/phytozome?organism=Ahypochondriacus, accessed on 1 August 2021). The following search parameters were set: full-tryptic protease specificity with two missed cleavages allowed, carbamidomethyl in cysteine (+57.021 Da), methionine oxidation (+15.995 Da), and asparagine/glutamine deamidation (+0.984 Da) as dynamic modifications, and precursor and fragment ion tolerances of ±10 ppm and ±0.1 Da, respectively, were applied. The resulting peptide hits were filtered for a maximum FDR of 1% using the Target Decoy PSM validator.

### 2.5. Protein Structure Prediction of Amaranth 11S Globulins

The 3D models of three Ah11sn paralogs identified in the LC–MS/MS analysis were performed with AlphaFold2 [12]. To model the 3D structures of the trimers, the models were subjected to an energy minimization step using the YASARA server [19] and then submitted to the GalaxyHomomer server [20]. A second energy minimization step was performed with the YASARA server. In addition, Procheck [21], Errat [22], and Verify3D [23] were used to assess the quality of the obtained models. The amino acids in disallowed regions were remodeled by the ModLoop server [24]. Electrostatic potential was calculated by the APBS-PDB2PQR software suite [25].

### 2.6. Dynamic Light Scattering (DLS) Analysis

DLS experiments were performed to study the oligomeric states of Ah11Sn as a function of pH, employing a Zetasizer µV (Malvern Instruments Ltd., Worcestershire, UK). Solutions of purified Ah11Sn (0.5 mg/mL) at pH 3, 6, 7, and 8 were filtered through syringe filters with a 0.22 µm pore size. The solutions were then immediately introduced into the DLS equipment. Hydrodynamic diameter (DH) vs. intensity percentage and molecular weight (Mr) determinations were analyzed with Zetasizer software version v3.30.

### 2.7. Surface Hydrophobicity

Surface hydrophobicity (H_0_) was determined as described previously [26]. Ah11Sn concentrations were calculated using 50 kDa, the average molecular weight of the Ah11Sn monomers [11]. Several Ah11Sn concentrations (200, 350, 500, 750, and 1000 nM) with a final concentration of 20 μM ANS were prepared. The effects of pH at 2 (50 mM Gly-HCl), 3–7 (25 mM citric acid and 50 mM Na_2_HPO_4_), and 8–9 (50 mM Tris-HCl) and the effect of NaCl (0–0.8 M, pH 3 and 8) on the H_0_ of Ah11Sn were also tested. The linear curve slope from the relative fluorescence unit (RFU) plots versus protein concentrations was defined as the H_0_ index [27]. The H_0_ index was reported versus pH or NaCl concentration. In addition, molecular docking analyses were performed to visualize the ANS binding sites using AutoDock Vina software (1.1.2, CCSB, La Jolla, CA, USA) [28].

### 2.8. Solubility

Protein solubility (S_0_) was determined according to a previously reported method [29] at different pH values (2–9) and NaCl concentrations (0–0.8 M) at pH 3 and 8. Protein dispersions of 0.2 mg/mL were prepared. The dispersions were vortexed for 2 min and then centrifuged (13,800× *g*, 20 min, 4 °C). The protein content in the supernatants was determined using the Bradford method [15]. S_0_ is expressed as the protein percentage content in the supernatant relative to the initial protein amount.

### 2.9. Emulsifying Properties

The emulsifying activity index (EAI) and the emulsion stability index (ESI) were determined according to the Pearce and Kinsella method [30] with some modifications. The preparation of o/w emulsions was as follows. Soybean oil (10%) was mixed with Ah11Sn dispersions (4 mg/mL) at pH 2–9 and NaCl (0–0.8 M, pH 3 and 8) using an Ultra Turrax T10 basic (IKA, Works, Inc., Wilmington, NC, USA) at 6000 rpm for 2 min. A 50-μL aliquot of the emulsion was taken from the bottom of the tube immediately after preparation and diluted in 0.1% (*w*/*v*) SDS at a ratio of 1:100 (*v*/*v*). Absorbance was measured at 500 nm after a 5 s vortex. The EAI was calculated using Equation (1):EAI (m^2^/g) = 2 × 2.303 × A_0_ × df/Cp × (1−θ) × 1000(1)
where df stands for dilution factor (100), Cp is the initial concentration of protein (g/mL), θ represents the fraction of oil used in the obtained emulsion (0.1), and A_0_ is the absorbance at 500 nm of the diluted emulsion. The ESI was expressed as the percentage of the emulsion content from the bottom of the glass at 0.25, 0.5, 1, 2, 4, and 24 h divided by the emulsion content of the initial solution.

### 2.10. Statistical Analysis

All tests were conducted in triplicate for individually prepared samples, and data are expressed as the mean ± standard deviation (SD). Where applicable, an analysis of variance (ANOVA) and multiple comparisons of means were carried out by applying the Tukey test (*p* < 0.05) using Stata MP software (16, StataCorp LLC, Lakeway Drive, TX, USA).

## 3. Results and Discussions

### 3.1. Purification and LC–MS Identification of Amaranth 11S Globulins

In this work, we report a novel two-step purification method for Ah11Sn. A first purification step with DEAE resin before SEC in S-300 resin allowed for cleaning the sample and provided high yields of purified protein (~10 mg/g of defatted flour) compared to those previously reported [11]. Using densitometric analysis, we observed a purity level of up to 95% (Figure 1B). The elution of crude protein extract on DEAE resin showed a significant peak corresponding to fractions 1–5 (Appendix A). After the second chromatography analysis on S-300 resin, the elution of 11S globulin showed one central peak, corresponding to fractions 59–69 (Appendix A).

Several oligomeric forms may be inferred for Ah11Sn based on their Mr calculated from the elution volume of 59–69 SEC fractions (Appendix A). Fractions 61–63 coincide with the standard protein marker of γ-globulin (158 kDa), which may indicate the presence of a significant population of trimers (~121.8–172.8 kDa), in addition to minor populations of tetramers in fractions 59–60 (~205.8–245.2 kDa) and dimers (85.8–102.2 kDa) and monomers (~42.6–72 kDa) in fractions 64–69. Similar behavior was reported in the S-300 elution profile of 11S globulin of chan seeds when observing the assembly from trimer to hexamer [31].

The electrophoretic profile of Ah11Sn under reducing conditions showed several protein bands (Figure 1A). The intense bands in the range of ~27–30 kDa and ~18–24 kDa corresponded to acidic and basic subunits of Ah11Sn and were named 11Sα1-2 and 11Sβ1-5, respectively. These results are consistent with the electrophoretic patterns of Ah11Sn reported previously [11] and revealed the presence of a highly conserved disulfide bond among 11S globulins. Densitometry analysis revealed that bands 11Sα1-2 (29.6 and 27.6 kDa) represented 32.7% of the total relative intensity, and basic subunits 11Sβ1-5 (23.6, 21.8, 20.4, 19.4 and 18.3 kDa) accounted for 61.2%, with 11Sβ3 being the most predominant band (Figure 1B). Additionally, two bands in the range of ~33–35 kDa corresponded to 7S globulins according to the results obtained by LC–MS/MS analysis and were named 7S1 (34.6 kDa) and 7S2 (33.6 kDa) (Appendix A). The presence of 7S globulins may be due to their similar solubility to saline-soluble 11S globulins, thus eluting with them in saline extraction buffers and remaining in the subsequent purification steps because of their structural similarity and equivalent Mr (~150–170 kDa) to the 11S globulin trimers [32,33]. The monomers of 7S and 11S globulins are structural homologs, so populations of 7S–11S heterotrimers and trimers consisting exclusively of 7S or 11S globulins may be viable [8].

LC–MS/MS analysis of the Ah11Sn bands (Appendix A) revealed the presence of complete and partial sequences of Ah11Sn paralogs. Two partial sequences (AH017743-RA and AH017744-RA) with theoretical Mr values of 13.2 and 32.7 kDa and three complete sequences named Ah11SB (AHYPO_001411-RA), Ah11SC (AH017742-RA), and Ah11SHMW (AHYPO_021282-RA) with theoretical masses of 55.4, 45.7, and 77.6 kDa, respectively, were identified.

All Ah11Sn monomers presented highly conserved structural features of 11S globulins containing two equivalent cupin β-barrel domains and the proteolytic site Asn-Gly cleaved by a specific asparaginyl endopeptidase generating the α and β subunits linked by a disulfide bond (Figure 2). However, some structural differences were observed; Ah11SB and Ah11SC have larger acidic chains and short basic chains, as previously reported for canonical Ah11SA (PDB ID 3QAC) [34]. In addition, Ah11SHMW is an 11S globulin paralog of high Mr, showing the largest acidic chain. Ah11SHMW also contains a C-terminal repeat domain (CTD) with several features, such as tandem repeats with conserved Ser and Tyr residues that could be involved in the signaling process by phosphorylation; His and Arg, positively charged amino acids that can have a significant impact on its functional properties, such as H_0_ and S_0_; and two Pro residues, an amino acid known for forming rigid points in secondary structures [35].

There was a significant difference between the theoretical Mr of the monomers of Ah11Sn and the experimental Mr values of its 11Sα and 11Sβ subunits, mainly in Ah11SHMW, whose theoretical Mr of 77.4 kDa was not equivalent to the sum of the experimental Mr of any of the 11Sα1-2 and 11Sβ1-4 subunits. This may be because 11S globulins are commonly subjected to posttranslational proteolysis mediated mainly by cysteine proteases, allowing the acquisition of their mature structure and proper packaging in seed vacuoles [36].

### 3.2. Assembly Capacity

11S globulins may undergo structural changes during food processing under different pH conditions. Their quaternary structure may change due to the formation of different oligomeric states that can modify the behavior of their physicochemical and functional properties [29,31].

To analyze the assembly capacity of Ah11Sn as a function of pH in the absence of NaCl, we calculated the D_H_ of the protein by DLS analysis. The DLS intensity size distribution and mass percentage of Ah11Sn (Figure 3) shows that, generally, at pH values of 3, 6, 7, and 8, the main population with a D_H_ of 10.2–10.7 nm (peak 1) predominates. The calculated Mr indicates the presence of mainly ~150–170 kDa oligomers, indicating that this protein may tend to dissociate, with trimeric assembly being predominant. These results are consistent with the relative Mr values of the principal peak fractions (62–64) in the S-300 elution profile (Appendix A), with relative molecular weights of ~121.8–172.8 kDa. Predominant trimeric oligomeric states have been reported for other 11S globulins, such as soybean glycinin [37], chan [31], and oat legumin [38].

Minor populations of high Mr were present in the sample of purified Ah11Sn. At pH 3, high Mr aggregates showed high-intensity peaks (2–3) where the mass percentage (% mass) of high Mr aggregates was lower than 1%; however, at pH 6, 7, and 8, high Mr aggregates populations are not statistically significant and undetected by equipment software (therefore % mass = 0), with respect to trimers populations with high intensity values and % mass of 100% (Figure 3B–D). Moreover, the mass percentage (% mass) of high Mr aggregates was lower than 1%, indicating a minor population in contrast with % mass up to 100% of peak 1. D_H_ values consistent with quaternary structures such as hexamers or dodecamers were not observed; this may be due to the possible dissociation of packaging hexamers or high molecular weight aggregates into trimers during the extraction steps. The association/dissociation behavior has been extensively studied in seed legumins, establishing that this phenomenon is mainly dependent on pH and ionic strength conditions [31].

DLS analysis is a valuable approach to study the predominant Ah11Sn quaternary structure under distinct pH conditions, which have a determinant influence on their functional and physicochemical properties and would greatly help predict their behavior and potential application in commercial or novel food systems.

### 3.3. Three-Dimensional Structure Prediction of Amaranth 11S Globulins

In amaranth seeds, only canonical Ah11Sn, one of the most abundant SSPs, has been characterized at the structural level by X-ray crystallography and named Ah11SA (PDB ID 3QAC) [34]. In this work, we generated the 3D structures of the three complete sequences of Ah11Sn paralogs named Ah11SB, Ah11SC, and Ah11SHMW, identified by LC–MS/MS analysis using AlphaFold2, which employs the DeepMind learning algorithm [12]. All Ah11Sn monomers were modeled as trimers (Figure 4) based on the results obtained by DLS analysis, which showed a predominant trimeric assembly.

The validation parameters of the generated models were evaluated based on Procheck, Errat, and Verify3D (Appendix A). Procheck provides a detailed evaluation of the stereochemistry of the conformation of the main chain. It generates a graph of conformational angles of each residue—φ angle (rotation around the N-Cα bond) and ψ angle (rotation around the Cα-C bond of the same Cα atom)—and a complete list of the residues. The results revealed that up to 88% of the residues of all Ah11Sn models were in favored regions, 8.6–11.1% were in allowed regions, 0.0–0.1% were in generous regions, and 0.0% were in disallowed regions. Errat verifies the structure of the protein by detecting local errors based on the statistics of unbound atomic interactions and comparing them with statistics of highly refined structures to suggest an overall quality factor. The results showed factors of 87.3–97.9%. Verify3D provided values of 64.3–82.5%. These results generally indicate good validation of the protein structures [21,22,23]. Therefore, the 3D structure models were compatible with the amino acid sequences. The Procheck, Verify3D, and Errat scores of the selected model were within acceptable ranges.

3D structure models of Ah11Sn paralogs generally showed structural features similar to those of canonical 11S globulins [33,39,40], with some particular characteristics (Figure 4). All Ah11Sn monomers, including the canonical Ah11SA, present two equivalent cupin domains containing the conserved β-barrel core and an extended α-helix domain containing several long and short helixes. These domains are superimposed by a pseudo-dyad symmetry with the N-terminal region oriented toward the intrachain disulfide bond face (IA face) and the C-terminal region toward the interchain disulfide bond face (IE face). Trimers are stabilized mainly by hydrophobic interactions through the extended α-helix domains, and the N-terminal region of one monomer associates with the C-terminal region of another in an asymmetric arrangement. The structural differences between Ah11Sn paralogs rely mainly on unstructured regions and loops. Ah11SA, Ah11SB, and Ah11SC have a higher distribution of extended α-helices with a triangular symmetry orientation on the IE and IA faces than Ah11SHMW, in which long unstructured regions oriented toward the IE face predominate. Appendix A shows the Cα backbone superimposition of Ah11SA with Ah11SB, Ah11SC, and Ah11SHMW. Ah11SA has root-mean-square deviations (RMSDs) of 1.23 Å compared with the 316 Cα atoms of Ah11SB, 1.92 Å for the 293 Cα atoms of Ah11SC, and 1.79 Å for the 355 Cα atoms of Ah11SHMW. The low RMSD values indicate that all globulins are structural homologs.

The surface charge distribution of proteins is very relevant since they dictate the behavior of their physicochemical and functional properties. The variations in the distribution of superficially charged residues can confer differentiated functional properties to each 11S globulin, such as S_0_ and H_0_, or the ability to form interactions with other molecules.

To calculate the surface electrostatic potential of Ah11Sn 3D models, we employed the APBS-PDB2PQR software suite [25]. The surface electrostatic potential maps showed that Ah11Sn paralogs are mainly negatively charged on their IA face and positively charged on their IE face (Figure 5). This feature is most notable in Ah11SA, Ah11SC, and Ah11SHMW, where the IA face is highly positively charged, in contrast to Ah11SB, whose IA face presents a positively and negatively charged zone distribution. These results are consistent with those reported for canonical Ah11SA [11] and other 11S globulins, such as soybean glycinin [39] and chickpea legumin [40], whose distribution of negatively and positively charged residues predominates on the IA and IE faces, respectively.

### 3.4. Surface Hydrophobicity

The H_0_ of proteins is related to physicochemical and functional properties such as S_0_ and emulsifying and foaming abilities [26]. For the experimental determination of H_0_, we employed a novel microvolumetric method using a Nanodrop fluorospectrometer and the fluorescent probe ANS, which binds to the surface of proteins through hydrophobic interactions with aromatic and aliphatic residues. This method has remarkable advantages over conventional methods since it reduces the protein and fluorophore quantities necessary for sample preparations and readings by two and three orders of magnitude, respectively [26].

The RFU_460_ of Ah11Sn was measured at pH 2–9 and NaCl 0–0.8 M within linear range concentrations on the order of nM (Appendix A). The values of each slope, defined as the H_0_ index, reflect the affinity of ANS toward the protein surface on each condition (Figure 6). The highest H_0_ index values as a function of the pH of Ah11Sn (Figure 6A) were obtained at pH 2 and 3 (3.2 and 3.5, respectively) and decreased as the pH increased, with the lowest value at pH 9. Remarkably, at acidic pH 2 and 3, the H_0_ of Ah11Sn increased up to 3.5, which could be related to the increase in exposed hydrophobic regions in the IE face due to the increase in positively charged surface regions (Figure 5) by the protonation of residues such as His, which could cause repulsion between the monomers. At pH 5, the H_0_ decreased more than 30% (2.1) compared with pH 3, while at pH 7–9, Ah11Sn showed less than 15% of the H_0_ at pH 3. Similar behavior has been previously reported with cruciferin from *Arabidopsis thaliana*; the wild-type (WT) cruciferin is a heterogeneous mixture of subunits contributed by different paralogous subunits named CRUA, CRUB, and CRUC [41]. The maximum H_0_ shown by WT cruciferin at pH 2 (2200) was more than five times higher than that shown at pH 7.4 (400), while CRUC, composed of identical subunits, showed a maximum H_0_ at pH 2 of 1600 and a minimum of 280 (pH 7.4), equivalent to 17% of the initial value.

The effect of different NaCl concentrations on the H_0_ index of Ah11Sn was measured at pH 3 and 8 (Figure 6B), at which the maximum and minimum H_0_ index values were obtained without NaCl, respectively. In general, adding NaCl had a negative effect on the H_0_ index at pH 3 and 8. At pH 3, adding 0.2 M NaCl resulted in a 30% decrease in the H_0_ index, reaching up to 54% with 0.8 M NaCl. Similarly, at pH 8 with a low H_0_ index (0.2), adding NaCl only further decreased it. The negative effect of NaCl on the H_0_ of Ah11Sn may be due to increased ionic interactions between NaCl and the positively and negatively charged surface regions of the protein.

To evaluate the formation of the Ah11Sn-ANS complex, molecular docking analyses were performed with AutoDock Vina software using the 3D models of Ah11SB, Ah11SC, and Ah11SHMW (Appendix A). The results showed interaction energies of −6.8 to −8.4 kcal/mol, which were within acceptable ranges [28]. In the Ah11SB-ANS complex, interactions of ANS with aliphatic residues such as Ala, Val, Leu, Ile, Met, and Pro and in smaller proportions with aromatic residues such as Phe, Tyr, and Trp were mostly observed. The Ah11SC/Ah11SHMW-ANS complexes mainly interacted with Phe and Tyr and, to a lesser degree, with Val and Ile.

### 3.5. Solubility

Protein S_0_ is an essential thermodynamic parameter related to the physicochemical and functional properties of proteins and thus may affect its application in food formulations [11]. The S_0_ of Ah11Sn was measured at pH values of 2–9 and 0–0.8 M NaCl.

The Ah11Sn S_0_ profile as a function of pH presented a U shape (Figure 7A). S_0_ increased to 98% at pH 2 and 3 (Figure 7A). At alkaline pH, S_0_ increases up to 80% at pH 8. The increase in S_0_ at acidic and alkaline pH can be explained by the polarity of the surface charge distribution in the IE and IA faces of the Ah11Sn trimers (Figure 5). At acidic pH 2 and 3, the protonation of residues in IA changed the distribution of positive surface charges; in contrast, at pH 8, the deprotonation of residues increased the IE face negative surface charges in all of these conditions, and the increase in hydrophilic interactions was favored, increasing the S_0_. An interesting behavior is observed at pH 9 since, in this condition, the solubility drops to ~40%, the presence of a high negative charge density in the dissolution environment could produce the exposition of hydrophobic surface patches on the IA and IE faces or conformational changes that promote the interaction amongst hydrophobic regions, thus decreasing the S_0_.

The lowest S_0_ value (8%) of Ah11Sn was presented at pH 4 and 5 without significant differences. The low solubility at these pH values can be explained by the neutral electrical charge of the proteins at their isoelectric points (pI). As a result, electrostatic repulsive interactions are not favored, promoting the formation of insoluble protein aggregates. The increase in solubility at pH values higher or lower than the pIs is related to electrostatic repulsive forces between positively or negatively charged proteins, which favor increased protein–solvent interactions [42]. A similar U-shaped S_0_ profile has been reported for canonical Ah11SA at low ionic strength (µ = 0.2) [11] and other 11S globulins, such as *A. thaliana* [41], breadnut [43], mungbean [32], fava bean, pea, and soybean [33].

The effect of NaCl concentrations on Ah11Sn S_0_ was evaluated at pH 3 and 8, at which the maximum S_0_ values were obtained (Figure 7B). Under both pH conditions, the addition of NaCl and the increase in its concentration have a negative effect on the S_0_ of Ah11Sn. At pH 3, S_0_ gradually decreased to 63.3% with an increment of 0.8 M NaCl. At pH 8, there was no significant difference (*p* < 0.05) with the addition of 0.1 and 0.4 M NaCl, but there was a decrease to 70% with 0.2 and 0.6 M NaCl and a decrease to 60.5% with 0.8 M NaCl. Under both pH conditions, the gradual increment of NaCl decreased the Ah11Sn S_0_. The negative effect of NaCl on S_0_ could be attributed to the neutralization of the positive and negative charges in the IA and IE faces due to the increase in ionic interactions, which decreases hydrophilic interactions. A similar result was observed for chan 11S globulin, whose solubility decreased with increasing concentration at 0.4 M NaCl [29].

### 3.6. Emulsifying Properties

The capacity of proteins to form and stabilize emulsions is a critical factor that determines their applications in the food industry as ingredients in food formulations. Proteins are hydrocolloids that can function as emulsifiers and stabilizers of oil-in-water (o/w) emulsions [43,44]. The EAI is a measurement of the interfacial area stabilized per unit weight of a protein (m^2^/g) and, thus, indicates the ability of a protein to coat an interfacial area [44]. The EAI of Ah11Sn was measured as a function of pH (2–9) and NaCl concentration (0–0.8 M) (Figure 8).

The highest EAI values as a function of pH for Ah11Sn (Figure 8A) were presented at pH 2 (191.06 m^2^/g) and 3 (130.8 m^2^/g), with significant differences (*p* < 0.05) when compared to other pH values. The next highest value for the EAI (113.1 m^2^/g) was observed at pH 7, followed by pH 6 (101.1 m^2^/g). No significant differences were observed at pH values of 4, 5, and 8 (*p* > 0.05). The lowest EAI value (54 m^2^/g) of Ah11Sn was presented at pH 9.

The higher EAI values at acidic pH values of 2–3 may be related to the equilibrium between H_0_ (3.2–3.5) and S_0_ (93.4%–98.7) at that same pH condition (Appendix A). The balance between the exposed hydrophilic and hydrophobic regions may allow the protein to establish more interactions between the o/w interface of the emulsion, acting as a suitable surfactant. Acidic pH conditions could cause partial unfolding of the Ah11Sn trimers, allowing them to expose hydrophobic regions and increasing the interfacial area between the o/w interface. These results differ from other reports of EAI of other 11S globulins, such as chan 11S globulins, whose highest EAI values were found at pH 9 [44], or breadnut globulins at pH 4 [43]. These differences may be mainly due to differences in charge distribution and hydrophobic residues, in addition to their assembly capacity.

The decrease in EAI at pH 4–5 is likely due to the pI of Ah11Sn, which prevents its action as a surfactant due to its neutral electrical charge favoring only hydrophobic interactions. At pH 7, a slight increase in EAI occurs, which can also be explained by the change in the charge distribution of the IA and IE faces, which favors a balance between hydrophilic and hydrophobic surface interactions. In contrast, at pH 8–9, hydrophobic interactions are more favored, changing this balance and decreasing EAI.

The addition of NaCl concentrations had a negative effect on the EAI values of Ah11Sn (Figure 8B), indicating that NaCl reduces electrostatic repulsion through ionic interactions. With the addition of 0.1 M NaCl, the highest EAI values were decreases of 42.6% (75 m^2^/g) and 25% (55.3 m^2^/g) at pH 3 and 8, respectively, compared to no NaCl addition. Increasing the NaCl concentration to pH 3 decreased the EAI values by 61% (50.9 m^2^/g) with 0.2 M NaCl compared to no NaCl addition and showed an increase of up to 81.5 m^2^/g with 0.8 M NaCl. At pH 8, the incorporation of 0.4–0.6 M NaCl showed no significant differences in the EAI values (*p* > 0.05) but showed a slight decrease compared with other NaCl concentrations.

The ESI is an additional index that measures the stability of a diluted emulsion over a fixed period and expresses the ability of a protein to form a stable emulsion without coalescence or flocculation [43,44]. The ESI values of Ah11Sn were determined at 0.25, 0.5, 1, 2, 4, and 24 h after emulsion formation (Figure 9). In the first 15 min, the highest ESI values of Ah11Sn were found at pH 8 (70.3%), followed by pH 4 (56.4%), pH 7 (51%), pH 6 (48.3%), pH 9 (44.7%), pH 2 (41.7%), pH 3 (40%), and pH 5 (12.7%). At 30 min under all pH conditions, the ESI of Ah11Sn decreased by more than 50%. The pH values with the best ESI values were 3 and 6, at which the ESI values were maintained at 40% and 36%, respectively, up to 1 h. The lowest ESI values were found at pH 5 and 4, decreasing up to ~88% in the first 15 and 30 min, respectively, due to closed values of the Ah11Sn pI, promoting the aggregation of the protein and avoiding its incorporation into the emulsion system.

In general, after 1 h, the Ah11Sn emulsions became unstable, decreasing by more than 40% and continuing to decrease until 24 h. Compared to the ESI values of other 11S globulins under different pH conditions, such as *A. thaliana* cruciferins [41] or soybean proglycinin [45], which are stable up to 20 h, Ah11Sn generally show unstable emulsions that decrease by more than 50% during the first hour. The EAI and ESI results indicate that Ah11Sn is suitable for potential applications as an emulsifier at acidic pH values in food formulations.

## 4. Conclusions

In this work, we present a two-step purification method for *A. hypochondriacus* saline-soluble 11S globulins suitable for an in-depth characterization of these proteins’ techno-functional and structural features. Ah11Sn is composed of three monomers named Ah11SB, A11SC, and Ah11SHMW; their 3D structures showed structural features similar to those of legumins, which generally consist of two equivalent cupin domains. Ah11Sn adopts trimeric quaternary structures, mainly negatively charged on their IA faces and positively charged on their IE faces; this bipolarity significantly affects its H_0_, S_0_, and emulsifying properties. Ah11Sn exhibited higher H_0_, S_0_, and EAI values at acidic pH, while S_0_ and ESI were well balanced at neutral pH. These results highlighted that Ah11Sn could be applied in formulating processed foods in the dairy, baking, and sausage industries. The perspectives of present work are the heterologous overexpression of each amaranth globulin 11S monomer in *Escherichia coli* to produce large quantities of single homogeneous monomer to assess its contribution on Ah11Sn physiochemical and functional properties and to carry out its food quality improvement by protein engineering. Finally, the different recombinant monomers could be used for enrichment of food systems.

## Figures and Tables

**Figure 1 foods-12-00461-f001:**
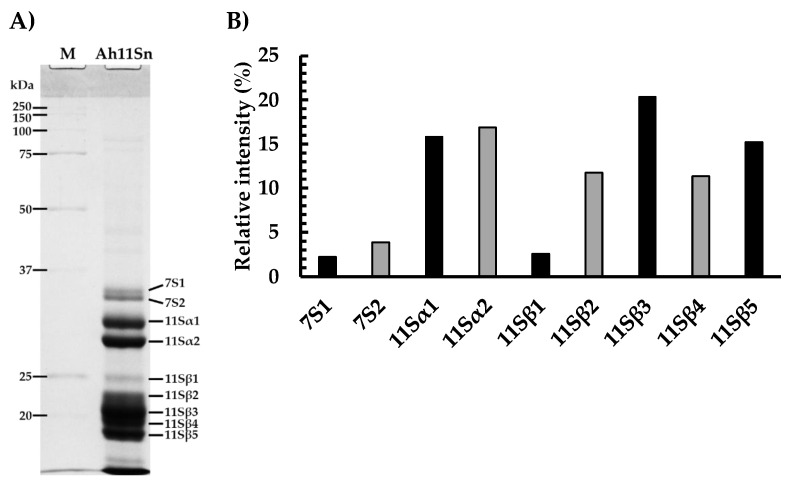
(**A**) Electrophoretic profile of Ah11Sn under reducing conditions. (**B**) Densitometry analysis.

**Figure 2 foods-12-00461-f002:**
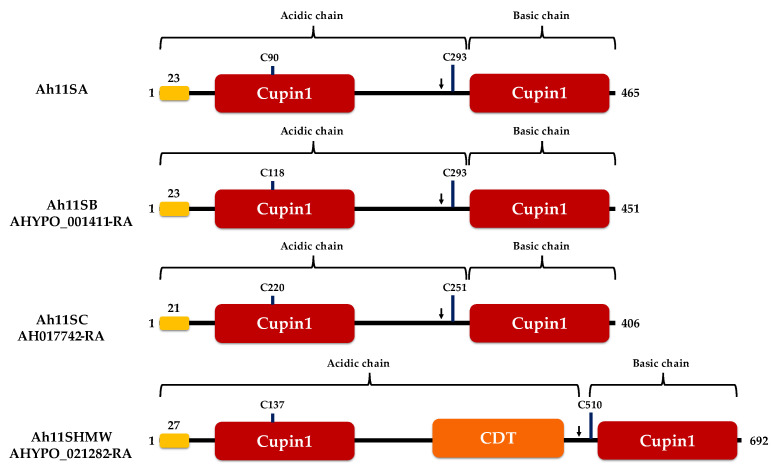
Conserved domains in amaranth 11S globulins. The arrow in each diagram indicates the proteolytic processing Asn-Gly site. The signal peptide is indicated in yellow. Cysteine residues forming the disulfide bond between the acidic and basic subunits are indicated.

**Figure 3 foods-12-00461-f003:**
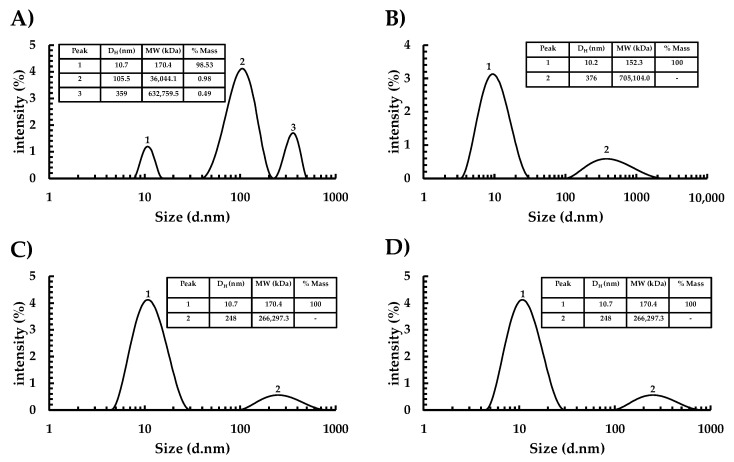
DLS intensity size distribution of Ah11Sn at different pH values. (**A**) pH 3, (**B**) pH 6, (**C**) pH 7, (**D**) pH 8. D_H_ and d correspond to hydrodynamic diameter, MW: calculated molecular weight in kDa.

**Figure 4 foods-12-00461-f004:**
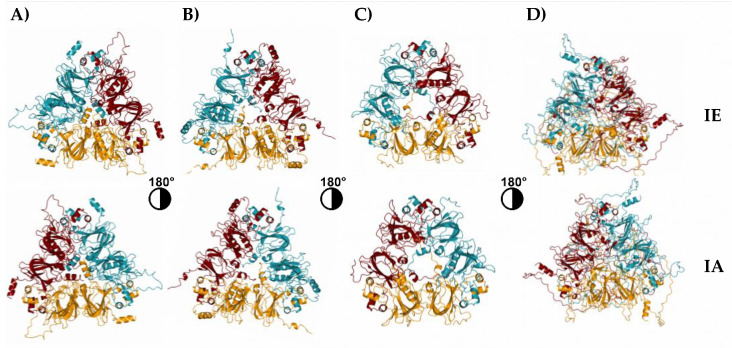
Structural models generated from Ah11Sn paralogs. (**A**) Ah11SA (PDB ID 3QAC); (**B**) Ah11SB; (**C**) Ah11SC; (**D**) Ah11SHMW. IA: intrachain disulfide bond face; IE: interchain disulfide bond face.

**Figure 5 foods-12-00461-f005:**
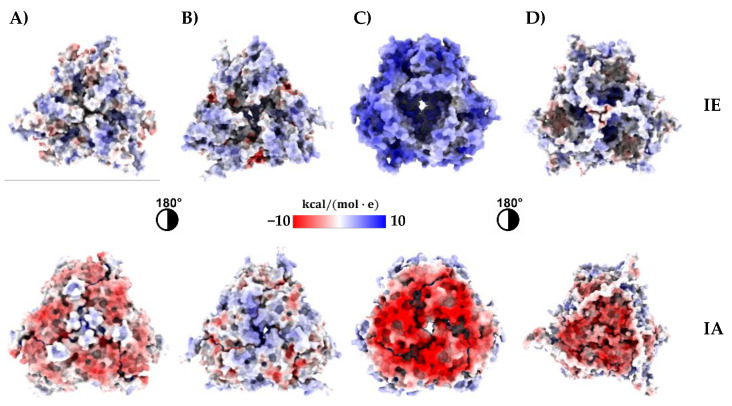
Surface electrostatic potential map of Ah11Sn paralogs. (**A**) Ah11SA (3QAC); (**B**) Ah11SB; (**C**) Ah11SC; (**D**) Ah11SHMW. IA: intrachain disulfide bond face; IE: interchain disulfide bond face.

**Figure 6 foods-12-00461-f006:**
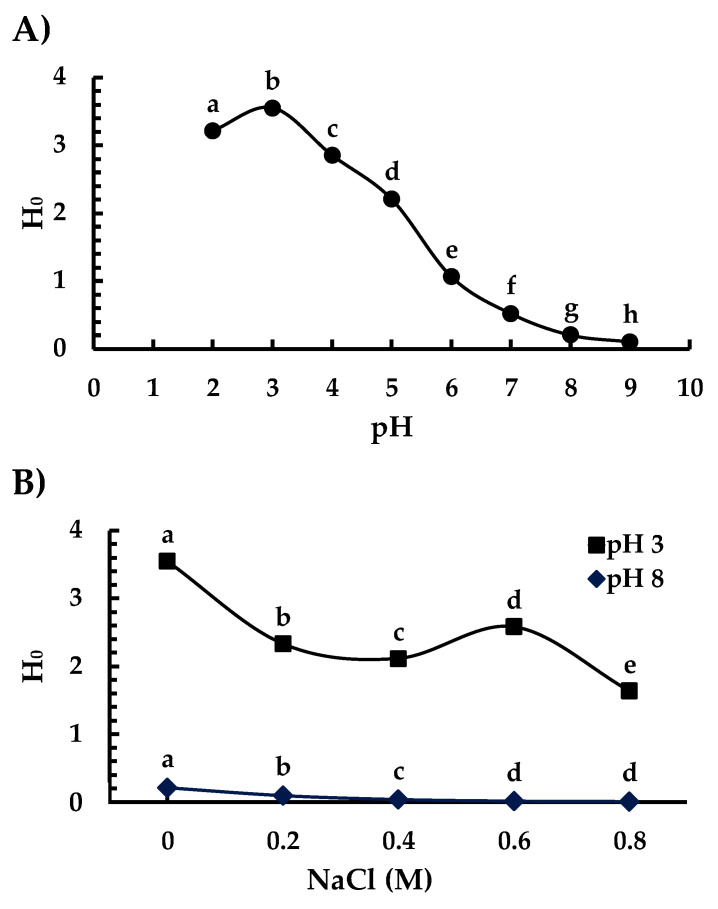
Surface hydrophobicity indices of Ah11Sn at different pH values (**A**) and concentrations of NaCl (**B**). The different letters at different pH values (**A**) or pH 3 or 8 (**B**) indicate significant differences at *p* < 0.05.

**Figure 7 foods-12-00461-f007:**
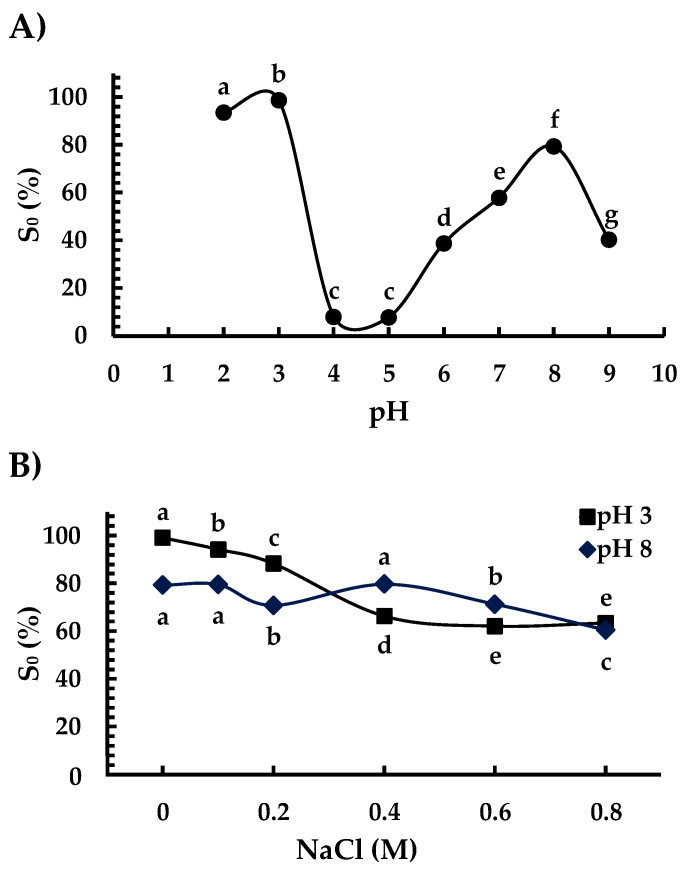
Solubility of Ah11Sn at different pH values (**A**) and concentrations of NaCl (**B**). The different letters at different pH values (**A**) or pH 3 or 8 (**B**) indicate significant differences at *p* < 0.05.

**Figure 8 foods-12-00461-f008:**
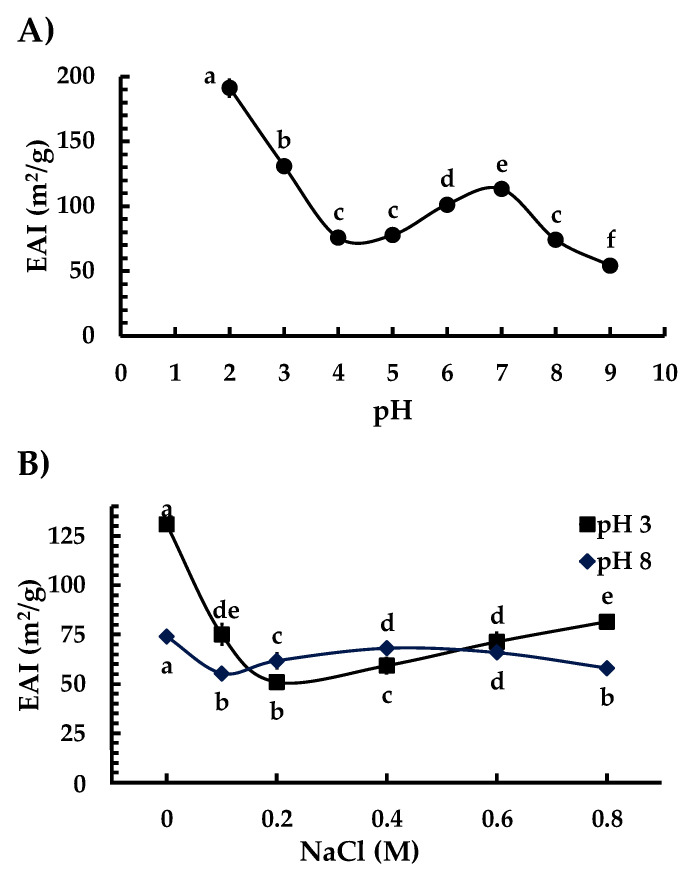
Emulsifying activity indices of Ah11Sn at different pH values (**A**) and concentrations of NaCl (**B**). The letters above or below the legends indicate significant differences at *p* < 0.05.

**Figure 9 foods-12-00461-f009:**
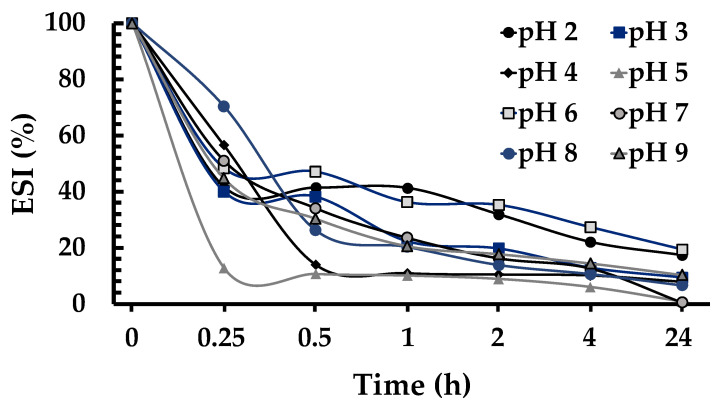
Emulsifying stability index of Ah11Sn.

## Data Availability

The data that support the findings of this study are available on request from the corresponding author.

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
