# Peer review of "Characterization of the Technofunctional Properties and Three-Dimensional Structure Prediction of 11S Globulins from Amaranth (Amaranthus hypochondriacus L.) Seeds"

_foods, 2023, doi:10.3390/foods12030461_

Round 1

Reviewer 1 Report

This manuscript addresses the characterization of the technofunctional properties and the prediction of the 3D-structure of 11S globulins from amaranth seeds. The authors aim to further explore the correlation between Ah11sn structure and function with the performance on different environmental conditions of interest in the food industry.

The paper is well written. However, the authors could further explore their conclusions by providing future perspectives based on the presented results, as well as a brief overview of the possible steps in the field.

Author Response

Answer:

Dear reviewer, the conclusions have been extended by adding aspects about the possible ways to continue exploring this protein and its applications in food science and technology.

Reviewer 2 Report

The manuscript studies the techno-functional properties of Amaranth 11S globulins at different pH conditions and NaCl concentrations. The work is interesting, well written, clear and carefully performed but needs some improvements before it can be accepted. Some results and explanation must be clarified in more detail. Here are several remarks and questions which have to be answered or took into account.

  1.       L202-203:Please confirm the molecular weight of bands 11Saα1-2 and 11Sβ1-5. For example, the bands of 11Sβ4-5 was lower than 15kDa, but the paper said the MW were 16.76 and 15 kDa, respectively.

2.       Fig.3B-D showed that some large aggregates existed, but the % Mass were zero. Please explain.

3.       The solubility at pH 9 was lower than at pH8, please explain.

4.       The EAI were lower at pH 8 and 9 than at pH 7, Why?

5.       The lowest ESI values were found at pH 5, please explain the reason.

Author Response

1. L202-203:Please confirm the molecular weight of bands 11Saα1-2 and 11Sβ1-5. For example, the bands of 11Sβ4-5 was lower than 15kDa, but the paper said the MW were 16.76 and 15 kDa, respectively. 

Answer:
Thank you very much for the observation, it was a mistake in the denomination of the bands in the marker and was corrected in the Figure, the MW of all bands were recalculated and corrected in the text and in the supplementary information. 

2. Fig.3B-D showed that some large aggregates existed, but the % Mass were zero. Please explain. 

Answer:
An explanation about this has been added: Minor populations of high Mr were present in the sample of purified Ah11Sn. At pH 3, high-Mr aggregates showed high-intensity peaks (2-3) where the mass percentage (% mass) of high-Mr aggregates was lower than 1%; however, at pH 6, 7, and 8 high-Mr aggregates populations are not statistically significant and un-detected for equipment soft-ware (therefore % mass=0), with respect to trimers populations with high-intensity values and % mass of 100% (Figure 3B-D).

3. The solubility at pH 9 was lower than at pH 8, please explain. 

Answer:
To clarify this, we add the following lines: An interesting behavior is observed at pH 9 since in this condition the solubility drops to ~40%, the presence of a high negative charge density in the dissolution environment could produce the exposition of hydrophobic surface patches on the IA and IE faces or conformational changes that promote the interaction amongst hydrophobic regions, thus decreasing the S0.

4. The EAI were lower at pH 8 and 9 than at pH 7, Why? 

Answer:
We propose a reason for this: The decrease in EAI at pH 4-5 is likely due to the pI of Ah11Sn, which prevents its action as a surfactant due to its neutral electrical charge favoring only hydrophobic interactions. At pH 7, a slight increase in EAI occurs, which can also be explained by the change in the charge distribution of the IA and IE faces, which favors a balance between hydrophilic and hydrophobic surface interactions. In contrast, at pH 8-9, hydrophobic interactions are more favored changing this balance and decreasing EAI.

5. The lowest ESI values were found at pH 5, please explain the reason.

Answer:
This has been explained: The lowest ESI values were found at pH 5 and 4, decreasing up to ~88% in the first 15 and 30 min, respectively, due to a closed value of the Ah11Sn pI, promoting the aggregation of the protein and avoiding its incorporation into the emulsion system.